# Onychomycosis in Two Populations with Different Socioeconomic Resources in an Urban Nucleus: A Cross-Sectional Study

**DOI:** 10.3390/jof8101003

**Published:** 2022-09-24

**Authors:** Pilar Alfageme-García, Víctor Manuel Jiménez-Cano, María del Valle Ramírez-Durán, Adela Gómez-Luque, Sonia Hidalgo-Ruiz, Belinda Basilio-Fernández

**Affiliations:** Department of Nursing, University Center of Plasencia, University of Extremadura, 10600 Plasencia, Spain

**Keywords:** onychomycosis, homeless, epidemiology, diagnosis, toenails

## Abstract

Onychomycosis is one of the most common foot conditions. Mixed onychomycosis and onychomycosis caused by non-dermatophyte moulds are increasing in incidence, especially in vulnerable populations, hence the importance of this study, which presents the prevalence of onychomycosis in a population of homeless people, comparing the findings with a sample of a well-resourced population. The total sample consisted of 70 participants, divided into two separate groups, a homeless population and a second group in which we included people attending a private clinic. The average age of the sample is [49.19 ± 28.81] with an age range of 18 to 78 years. In the homeless group, the most prevalent infectious agents were non-dermatophyte fungi, with a total of 48%, compared to 28% in the group housed. The most common site of infection in both groups was the nail of the first finger. We, therefore, conclude that there is a difference in the infecting agent in the homeless population and the population with homes.

## 1. Introduction

One of the most common infections that all podiatrists are confronted with on a daily basis is onychomycosis. It is estimated that 50% of nail diseases are fungal infections [1]. The main family that triggers these nail lesions is the dermatophytes. Incidence varies according to the geographical area and study population with figures such as 23% in Europe, 20% in East Asia and 14% in North America. In the USA, there are an estimated 10 million people with onychomycosis [2]. Approximately 10% of the general population suffers from these infections, rising to 20–30% in patients over 60 years of age and increasing to 50% in patients over the age of 70 [3,4].

The first toenail is the most frequently affected one [5,6]. Changing lifestyles, advanced age, obesity and immunosuppressed states such as diabetes mellitus, organ transplantation, corticosteroid use and antineoplastic drugs are increasing the prevalence of yeast and non-dermatophyte fungal infections [5,7].

Onychomycosis can limit daily living activities, sometimes causing local pain and paraesthesia, making daily living activities more challenging. It is also associated with a negative psychosocial impact that may cause patients to become socially isolated [8].

Given the high prevalence of this chronic infection in podiatric clinics, it seems important to continue research in different population sectors in order to establish accurate diagnoses regarding the relationship between the pathogen causing the infection and the conditions of the host who shows signs and symptoms and suffers from onychomycosis. In this research, we describe two types of populations. The first to be studied is a group of homeless people, whose definition in the social framework is ambiguous given that it is a heterogeneous group of individuals. We can make an approximation with the definition provided by the Federation of National Associations that Work in Favour of the Homeless (FEANTSA), understanding homeless people as individuals who are unable to access and maintain personal and adequate accommodation by their own means or with the help of social services, as well as those people who live in institutions (hospitals, prisons, social services, etc.) but have no personal accommodation to go to when they leave and people living in subhuman or clearly overcrowded accommodations [9]. The second study group, as far as fungal infections is concerned, is people with a roof over their heads and a stable socioeconomic situation, who demand foot care within the private health care system.

This work has been carried out in Plasencia, a city of inhabitants [10] located in the north of Extremadura, within the collaboration agreement of Integral Health Care for the Homeless, belonging to Cáritas, in which our research group GICISA, included in the catalogue of groups of the University of Extremadura, provides podiatric care to this sector of the population, cooperating with the podiatric clinic “on the street” in Plasencia.

In this article, we describe aspects related to the prevalence of onychomycosis in a population of homeless people, where doctors attend a shelter to provide care versus patients visiting a podiatry clinic in a town in the North of Cáceres (Spain). Populations at increased risk of infection are described and the prevalence of onychomycosis reported in the literature has not yet been summarised in these risk groups. We performed a systematic review of the literature and calculated pooled prevalence estimates of onychomycosis in at-risk patient populations. There are predisposing factors, which are specific to different population sectors, including older age, tinea pedis, immunosuppression, or male sex, for developing a fungal nail infection [11].

For this reason, we set out to compare how fungal infection affects two populations with different socioeconomic resources (housed and homeless) to determine whether there are significant differences in the type of agent that causes nail disease and its possible causes.

The purpose of this study is to isolate and identify onychomycosis-producing agents in homeless people with clinical suspicion of fungal infection of the toenails and compare them with agents isolated from populations attending podiatric clinics with clinical suspicion.

## 2. Materials and Methods

### 2.1. Design, Patients and Sample Collection

A cross-sectional descriptive study was conducted in 70 selected patients.

Participants were included in the study when meeting the following selection criteria: age ≥ 18 years, clinical suspicion of fungal infection on toenails, willingness to give informed consent and ability to be present at the time of sample collection.

Our sample was a convenience sample. The sample collection consisted of taking nail fragments from the 70 patients with clinical suspicion of fungal infection on toenails, following anamnesis and clinical examination of the nail plate. A single researcher collected all samples.

There were two settings. In one, patients were selected at the consultation room set up for health care at the Cáritas social health centre for the homeless located in Plasencia (*n* = 30). In the other, patients were selected at the chiropody clinic “on the street” in Plasencia (*n* = 40). Both facilities are located in the north of Extremadura (Spain).

A checklist was created for data collection, including socio-demographic data and variables to be studied in the investigation. The variables recorded were age, sex, overweight (BMI ≥ 25), diabetes mellitus (DM), physical activity (30 min each day or 4 h per week), immunosuppressive medications or conditions such as Hepatitis B or HIV, clinical suspicion of fungal infection on the toenails, location of onychomycosis and infecting agent.

### 2.2. Protocol for Collected Samples

To reduce the risk of contamination during sample collection, a “sampling protocol” was used. In addition, both environments were sterilized in the same way prior to sample collection using ozone.

The first step was to clean and disinfect the nail plate with 70° alcohol, taking the fraction of the toenail most suitable for cultivation after the antiseptic had evaporated, taking of the active part of the lesion [5].

### 2.3. Clinical Diagnosis

The clinical diagnosis consisted of a visual examination, determining the location of the lesions on the toenails [8].

We classified the clinical presentation as:DSO: distal subungual onychomycosis.DLSO: disto-lateral subungual onychomycosis.TDO: total dystrophic onychomycosis.MO: mixed onychomycosis, presenting characteristics of more than one pattern of infection.

There are some pictures available as Appendix A.

### 2.4. Isolation of the Infecting Agent

The infectious agent was isolated in the microbiology department of the Adame-Eurofins Megalab clinical analysis laboratory in Plasencia.

### 2.5. Ethical Standards

This study was conducted following the guidelines of the Declaration of Helsinki, whilst the handling of human samples was approved by the Bioethical Commission of the Universidad de Extremadura (ref 85/2022). Before the collection of samples, all the participants gave their verbal and written consent.

### 2.6. Statistical Analysis

Statistical evaluation of the results was performed with the statistical package SPSS v.28.0.1 (SPSS Chicago, IL, USA) for Windows.

The statistical method included describing the participants’ demographic and clinical characteristics expressing them as absolute values, mean ± SD or as percentages. In addition, univariate chi-square analyses were used to explore the relations between group population and fungal family and the location of lesions and clinical presentation. Finally, odds ratios with their 95% confidence intervals were calculated to determine the association between having positive culture and age, overweight, DM, physical activity and immunosuppressive medications or conditions. Statistical significance was considered when *p* < 0.05. 

## 3. Results

### 3.1. Sampling and Study Groups

In our study, we had a total sample of 70 individuals whose average age was [49.19 ± 28.81] (with an age range of 18 to 78 years). In terms of sex, 25.7% (*n* = 18) of our total sample was female and 74.3% (*n* = 52) male (Table 1).

Of the total sample, 40 people belonged to the housed group compared to 30 who belonged to the homeless group.

In the housed group, participants’ average age was [48.05 ± 29.95] while the average age in the homeless group was [50.70 ± 23.7].

According to sex, 35% of the participants in the housed group were female compared to 65% male, while in the homeless group, 13.3% were female and 86.7% male.

Of the total number of analysed samples (*n* = 70), 50 positives cultures were identified, representing a 71.4% prevalence of onychomycosis.

In the housed group, 62.5% of the cultures were positive, i.e., 25 patients, of whom 9 were female and 16 were male. On the contrary, and even though the number of homeless people with onychomycosis was the same (25), the prevalence of infection was higher in the homeless group at 83.3%, with 3 females and 22 males infected.

### 3.2. Results after Positive Culture, Relationship between Groups of Fungi and Infectious Agents with Study Groups

The most frequent fungi group among the housed population was the dermatophyte, *Trichophyton rubrum* being the most encountered (40%, *n* = 11). Subsequently, the most frequent infectious agents are those belonging to the group of non-dermatophyte fungi, (28%, *n* = 7), the genus *Aspergillus* being the most frequent.

In the group of homeless people, fungi groups are inverted, with non-dermatophyte being the most common (48%, *n* = 12) and the *Aspergillus* (12%, *n* = 3) and *Acremonium* (12%, *n* = 3) the genero most encountered. From the dermatophyte group, the most prevalent species were *Trichophyton rubrum* (8%, *n* = 2) and *Trichophyton tonsurans* (8%, *n* = 2).

Mixed infections were more prevalent in this population group at 12% (*n* = 3) when compared to the 4% (*n* = 1) in the housed group.

Regarding yeasts, the most frequent infectious agent in both groups was *Candida parapsilosis*.

However, no statistical significance was found between fungal groups and our study groups (*p* = 0.36), as shown in Table 2.

### 3.3. Relationship between the Location of Onychomycosis and the Study Groups

Although there are no significant differences between the location of the mycotic lesions and the population groups studied (*p* = 0.15), as can be seen in Table 3, in the housed group, the most frequent location of the mycotic lesions is in the first toenail (52%, *n* = 13). On the contrary, in the homeless group, lesions affecting more than one toenail are more frequent (66.7%, *n* = 16).

### 3.4. Relationship between Clinical Presentation of Fungal Lesions and Study Groups

Regarding the general clinical presentation, statistically significant differences were found between clinical presentation and our study groups (*p* = 0.03). As shown in Table 4, in the housed group, total dystrophic onychomycosis (TDO) (28%, *n* = 7) and mixed onychomycosis (MO) (28%, *n* = 7) are more frequent. However, in the homeless group, distal subungual onychomycosis (DSO) (41.7%, *n* = 10) and mixed onychomycosis (MO) (41.7%, *n* = 10) are more frequent.

### 3.5. Associated Risks Related to Positive Culture and Socioeconomic and Confounding Variables

Being homeless was associated with having more risk to encounter a positive culture (X2 = 3.6; *p* = 0.05), (OR = 3.0; IC 95% = 1.0–9.5). In our sample, having diabetes mellitus created no greater risk of encountering a positive culture (*p* = 0.78) (OR = 1.7; IC 95% = 0.2–16.1), nor did immunosuppression (*p* = 0.64) (OR = 3.0; IC 95% = 0.8–11.7), overweight (*p* = 0.63) (OR = 2.7; IC 95% = 0.8–8.4), being physically active (*p* = 0.06) (OR = 1.2; IC 95% = 0.4–3.5) or being older than 50 years (0.33) (OR = 1.0; IC 95% = 0.3–2.7).

## 4. Discussion

Following this satisfactory research, we can say that we have not found any previous studies or publications that refer to the relationship between homelessness and the occurrence of onychomycosis of the foot.

According to our findings, the most common fungi group causing onychomycosis in our sample is dermatophytes, the genus *Trichophyton* being the most frequent, specifically the species *Trichophyton rubrum.* These findings are in consonance with previous studies, which stated that the main species causing onychomycosis are *Trichophyton rubrum* and *Trichophyton mentagrophytes*, these being the most prevalent worldwide [11,12,13].

When focusing on the two analysed groups in this study, there are evidenced differences in terms of the group of fungi and the infecting agents, non-dermatophytes being the most frequent among the homeless, specifically *Aspergillus* and *Acremonium*. However, in the housed population, dermatophytes of the genus *Trichophyton,* and in particular *Trichophyton rubrum,* are more frequent. Thus, despite the fact that our results are not statistically significant, previous studies by Fatahinia et al., 2010, Effendy, I. et al., 2005, Gelotar, P. et al., 2013 and Martinez-Herrera, E.O. et al., 2015, show that in the general population, that is, with socioeconomic resources, non-dermatophyte fungi are less frequent [14,15,16,17].

This fact leads us to think that several socioeconomic conditions have an influence on the appearance of fungal infections on toenails, determining the infecting group of fungi and prevalent infectious agents. The homeless population studied is characterized by a lack of socioeconomic resources and family coverage, as well as diseases and conditions linked to immunosuppression, such as HIV and hepatitis C.

In our research, we found that 63.3% (*n* = 19) of the homeless group had addictions, 30% were diagnosed with HIV (*n* = 9), 6.6% had Hep C (*n* = 2) and 60% (*n* = 18) of the individuals in this study population were immunocompromised compared to 5% of the housed population (*n* = 5). Despite the high percentage of immunocompromised individuals in the homeless population, we have not found this condition to be a predisposing factor for developing onychomycosis of the foot; in our case, we associated it with unhealthy lifestyles. However, there are authors who associate the occurrence of non-dermatophyte onychomycosis with both lifestyle and immunosuppression, as reflected in articles by Bongomin et al. and Lipner et al., with a particular increase in *Aspergillus* infections [7,8].

On this point, the Aspergillus infecting agent was the most common of the non-dermatophytes genus in both our groups. Furthermore, among the housed group, non-dermatophyte fungi is the second most frequent genus, these data being in accordance with Marco-Tejedor et al. who stated that there is a changing pattern of infecting agents, demonstrated by the increasing prevalence of non-dermatophyte-caused onychomycosis among the general population from rural areas [5].

Focusing on mixed infections is more common in our homeless population. According to the literature, mixed infections are frequently caused by a dermatophyte working as the infecting agent and a non-dermatophyte as the contaminating agent [18,19]. However, in our study, these data differ from the aforementioned bibliography, being explained and attributed to the hygiene conditions and lifestyle choices of the studied population.

In our study group, we have isolated onychomycosis in homeless people, with dermatological infections caused by fungi and bacteria; for this reason, we have not ruled out the possibility that the nail lesions were caused by fungi or even bacteria, although we have not taken this into account in our study. For future research, it is a factor to be taken into account since 46.6% of the sample (in homeless people *n* = 14) presented lesions not only in the nails but also in the skin, these dermatopathies related to bacteria and fungi.

As for the clinical suspicion of mycosis in the total study sample (*n* = 70), the diagnostic confirmation of mycosis by culture and microscopy was 71.4% of the total (*n* = 50). Regarding the housed group, this prevalence is 62.5%, which enhances the results obtained by Babayani, M. et al., in their study published in 2018 which found prevalence was 65.5% (*n* = 118) [20]. Aragón-Sánchez, J. et al., in their study published in 2021 using similar techniques as those used in this study, obtained diagnoses confirmation in 40.5% of the cases with clinical suspicion (*n* = 101) [21]. However, this comparison should be taken cautiously since their target population was only diabetic.

Regarding the toenail anatomical area affected by onychomycosis, our results are in consonance with a recent review carried out in 2022, the first toenail being the most affected area [6].

With regard to clinical presentation, TDO and MO are the most common among the housed population and among the homeless population DSO and MO. Notwithstanding that fact, we have not found any recent studies that match the clinical presentations of our research since most studies describe that the most frequent clinical presentation is DLSO [6,21,22]. This could be explained by our sample size.

In this study, we found no relationship between being diabetic, physically active, overweight, having immunosuppression or being over 50 years old with the appearance of fungal infections. However, most of the reviewed authors suggest that factors such as immunosuppression or age are related to the appearance of mycosis [5,7,8]. We believe that our results could change with an increased sample size or by taking into account various factors among the homeless population, such as poor hygiene habits, wearing inappropriate footwear and socks or unhealthy lifestyle habits, given that no research has taken into account these variables in the homeless population in previous studies. We are considering this as a future line of research for further work.

### Limitations

This study has some mentionable limitations. The main limitation is the sample size, which makes it difficult to generalize the results to the rest of the population; however, we could consider this study a first approach to this vulnerable population. It is in the researcher’s mind to continue working with this population, hence being able to increase the samples.

Another important limitation is not having taken into account variables that may explain the appearance of onychomycosis, such as diet, foot hygiene, footwear, etc.

A further limitation is that we have not considered the possibility of mixed fungal and bacterial infection, which would be a finding to be considered in future studies.

## 5. Conclusions

In our research, after studying two populations with different socioeconomic conditions, we can conclude that in homeless people, non-dermatophytes were the most prevalent fungi group, the genre *Aspergillus* and *Acremonium* being the most frequent. However, dermatophytes were the most prevalent fungi group in the housed population, the genus *Trichophyton* being the most frequent.

This may be due to lifestyle, hygienic and dietary habits and immunosuppression, which could be related to this type of fungal infection in our study population. Further research should be done in this type of population.

## Figures and Tables

**Table 1 jof-08-01003-t001:** Sociodemographic characteristics, confounding variables and identified infectious agents (*n* = 70).

Groups	Subjects	Years	Sex	Culture	Overweight	DM	Immuno-suppression	Physical Activity	Infecting Agents
Housed (*n* = 40)	1	54	Male	+	No	No	No	No	*Candida parapsilosis*
	2	41	Male	+	No	No	No	No	*Candida albicans*
	3	41	Female	+	No	No	No	Yes	*Trichophyton rubrum*
	4	37	Male	+	No	No	No	Yes	*Trichophyton rubrum*
	5	65	Female	+	No	No	No	Yes	*Trichophyton rubrum*
	6	53	Female	+	No	No	No	No	*Trichophyton rubrum*
	7	69	Male	+	Yes	No	No	No	*Aspergillus niger*
	8	37	Male	+	No	No	No	No	*Rhodotorula* spp. + *Exophila* spp.
	9	61	Male	+	Yes	No	No	No	*Candida parapsilosis + Rhinocladiella* spp.
	10	35	Male	+	No	No	No	Yes	*Aspergillus terreus*
	11	51	Female	+	No	No	No	No	*Aspergillus* spp. + *Cladosporium* spp.
	12	54	Male	+	Yes	No	No	No	*Candida parapsilosis*
	13	18	Male	+	No	No	No	No	*Trichophyton rubrum*
	14	50	Female	+	No	No	No	Yes	*Acremonium* spp.
	15	64	Female	+	Yes	No	No	Yes	*Aspergillus glaucus*
	16	49	Male	+	Yes	No	No	Yes	*Candida parapsilosis*
	17	64	Male	+	No	No	No	Yes	*Trichophyton rubrum*
	18	42	Male	+	Yes	No	No	No	*Aspergillus fumigatus*
	19	53	Female	+	Yes	No	No	No	*Trichophyton rubrum*
	20	63	Female	+	No	No	Yes	Yes	*Trichophyton rubrum*
	21	66	Male	+	Yes	Yes	No	No	*Trichophyton rubrum*
	22	32	Male	+	No	No	No	Yes	*Trichophyton rubrum*
	23	52	Male	+	Yes	Yes	No	No	*Candida parapsilosis*
	24	27	Female	+	No	No	No	Yes	*Trichophyton mentagrophytes*
	25	49	Male	+	No	No	No	Yes	*Mycospoum ferrugineum*
	26	45	Male	-	No	No	No	Yes	No
	27	56	Male	-	Yes	No	No	No	No
	28	43	Male	-	No	No	No	No	No
	29	23	Female	-	No	No	No	Yes	No
	30	48	Female	-	No	No	No	No	No
	31	48	Female	-	No	No	No	No	No
	32	42	Female	-	Yes	No	No	Yes	No
	33	49	Male	-	No	No	No	No	No
	34	47	Male	-	No	No	No	No	No
	35	53	Male	-	No	No	No	No	No
	36	21	Male	-	No	No	No	Yes	No
	37	49	Male	-	Yes	No	No	No	No
	38	45	Male	-	No	No	No	Yes	No
	39	48	Female	-	No	No	No	No	No
	40	78	Male	-	Yes	No	No	No	No
Homeless (*n* = 30)	41	48	Male	+	Yes	No	Yes	No	*Trichophyton rubrum*
	42	63	Male	-	No	Yes	No	No	No
	43	61	Male	+	No	No	Yes	No	*Acremonium* spp.
	44	45	Male	+	Yes	No	Yes	No	*Aspergillus* spp.
	45	42	Male	-	Yes	No	Yes	No	No
	46	55	Male	+	Yes	No	Yes	No	*Mucor* spp.
	47	61	Female	+	No	No	No	No	*Cladosporium* spp.
	48	55	Male	+	Yes	No	Yes	No	*Candida parapsilosis*
	49	43	Male	-	No	No	Yes	No	No
	50	57	Male	+	Yes	No	Yes	No	*Trichophyton tonsurans*
	51	63	Male	+	No	No	Yes	No	*Candida albicans*
	52	64	Female	+	No	No	No	No	*Aspergillus fumigatus*
	53	56	Male	+	Yes	No	Yes	No	*Trichophyton violaceum + Trichophyton schoenieinii*
	54	53	Male	+	Yes	No	Yes	No	*Fusarium* spp.
	55	37	Male	+	No	No	Yes	No	*Cladosporium* spp.
	56	49	Female	+	Yes	No	Yes	No	*Acremonium* spp.
	57	62	Male	+	No	No	No	Yes	*Trichophyton rubrum*
	58	48	Male	+	Yes	No	Yes	No	*Scytalidium* spp. + *Onychocola canadensis*
	59	51	Male	+	No	No	No	No	*Candida parapsilosis*
	60	39	Male	-	No	No	Yes	Yes	NO
	61	41	Female	-	No	No	No	Yes	NO
	62	39	Male	+	No	No	Yes	No	*Trichophyton Scholenieinii + Aspergilllus* spp.
	63	41	Male	+	No	No	No	Yes	*Fusarium* spp.
	64	54	Male	+	No	Yes	No	Yes	*Candida parapsilosis*
	65	49	Male	+	Yes	No	No	No	*Trichophyton Scholenieinii + Aspergilllus* spp.
	66	54	Male	+	Yes	No	No	No	*Rhodotorula* spp. + *Candida parapsilosis*
	67	46	Male	+	Yes	No	No	Yes	*Trichophyton Scholenieinii*
	68	49	Male	+	No	No	Yes	Yes	*Trichophyton tonsurans*
	69	54	Male	+	Yes	No	No	Yes	*Aspergillus terreus*
	70	42	Male	+	Yes	No	Yes	Yes	*Acremonium* spp.
	**Mean**	49.19 ± 28.81							

**Table 2 jof-08-01003-t002:** Relationship between groups of fungi and infectious agents with study group (*n* = 50).

Groups	Fungal Groups % (*n*)	Infecting Agents	% (*n*)	* *p*-Value
Housed (*n* = 25)	Yeast 24% (6)	*Candida parapsilosis*	16% (4)	0.36
		*Candida albicans*	4% (1)	
		*Rhodotorula* spp. + *Exophiala* spp.	4% (1)	
	**Dermatophyte** **44% (11)**	*Trichophyton mentagrophytes*	4% (1)	
		*Trichophyton rubrum*	40% (10)	
	Non-dermatophytes 28% (7)	*Aspergillus terreus*	4% (1)	
		*Aspergillus* spp. + *Cladosporium* spp.	4% (1)	
		*Acremonium* spp.	4% (1)	
		*Aspergillus glaucus*	4% (1)	
		*Aspergillus fumigatus*	4% (1)	
		*Aspergillus niger*	4% (1)	
		*Mycospoum ferrugineum*	4% (1)	
	Mixed 4% (1)	*Candida parapsilosis + Rhinocladiella* spp.	4% (1)	
Homeless (*n* = 25)	Yeast 16% (4)	*Candida parapsilosis*	12% (3)	
		*Candida albicans*	4% (1)	
	Dermatophyte 24% (6)	*Trichophyton rubrum*	8% (2)	
		*Trichophyton tonsurans*	8% (2)	
		*Trichophyton violaceum + Trichophyton schoenieinii*	4% (1)	
		*Trichophyton Scholenieinii*	4% (1)	
	**Non-Dermatophytes** **48% (12)**	*Cladosporium* spp.	8% (2)	
		*Aspergillus fumigatus*	4% (1)	
		*Aspergillus terreus*	4% (1)	
		*Aspergillus* spp.	4% (1)	
		*Mucor* spp.	4% (1)	
		*Acremonium* spp.	12% (3)	
		*Fusarium* spp.	8% (2)	
		*Scytalidium* spp. + *Onychocolac canadensis*	4% (1)	
	Mixed	*Trichophyton Scholenieinii + Aspergilllus* spp.	8% (2)	
	12% (3)	*Rhodotorula* spp. + *Candida parapsilosis*	4% (1)	

* ***p*-value**: statistical significance at a *p*-value < 0.05.

**Table 3 jof-08-01003-t003:** Results of relationship between mycotic lesion location and study group (*n* = 50).

Groups	* Localization of Lesions	% (*n*)	* *p*-Value
Housed (*n* = 25)	First toenail	52% (13)	0.15
	More than 1 toenail	48% (12)	
Homeless (*n* = 25)	First toenail	33.3% (8)	
	More than 1 toenail	66.7% (16)	

* ***p*-value**: statistical significance at a *p*-value < 0.05.

**Table 4 jof-08-01003-t004:** Results of relationship between clinical presentation of fungal lesions and study group (*n* = 50).

Groups	* Clinical Presentation	% (*n*)	** *p*-Value
Housed (*n* = 25)	DSO	24% (6)	0.03
	DLSO	20% (5)	
	TDO	28% (7)	
	MO	28% (7)	
Homeless (*n* = 25)	DSO	41.7% (10)	
	DLSO	16.7% (5)	
	MO	41.7% (10)	

* **DSO**: distal subungual onychomycosis; **DLSO**: disto-lateral subungual onychomycosis; **TDO**: total dystrophic onychomycosis; **MO**: mixed onychomycosis, with characteristics of more than one pattern of infection. **** *p*-value**: statistical significance at a *p*-value < 0.05.

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
