# Peer review of "Onychomycosis in Two Populations with Different Socioeconomic Resources in an Urban Nucleus: A Cross-Sectional Study"

_jof, 2022, doi:10.3390/jof8101003_

Round 1

Reviewer 1 Report (Previous Reviewer 2)

In this study is to isolate and identify onychomycosis‐etiologic agents in homeless people with clinical suspicion of fungal infection of the toenails and compare them with agents isolated from populations attending podiatric clinics with clinical suspicion. The manuscript is interesting and has improved since its previous submission. I have some comments.

In Statistical analysis section does not describe regression analyzes (OR). 

Results. 

Tables 2, 3 and 4. I suggest remove X2

Associated risks related to positive culture and socioeconomic and confounding variables   

It is not clear whether the analyzes are crude or adjusted.

Also in the OR you must include the p-value

Several paragraphs in this section are repetitive of the Results section.

I suggest to improve the discussion

Consider the title "Onychomycosis in two populations with different socioeconomic resources in an urban nucleus: A Cross-sectional study". 

Author Response

September, 16th 2022

Dear Reviewers

Please, find attached a revision of our manuscript entitled “Onychomycosis in two populations with different socio-economic resources in an urban nucleus: A Cross-sectional study”.  We would like to thank the reviewers’ comments. We have considered all suggestions and included them in the revised manuscript. We believe that our manuscript is stronger as a result of these modifications. You will find the changes in blue and red colour in the resubmitted manuscript. We hope that the manuscript may be of interest to Journal of Fungi.

Regards,

Prof. Adela Gómez Luque

Universidad de Extremadura

Reviewer #1

Thank you very much for your contribution and suggestions.

In Statistical analysis section does not describe regression analyzes (OR). 

Corrected in the text

Associated risks related to positive culture and socioeconomic and confounding variables   

It is not clear whether the analyzes are crude or adjusted.

The analyses are crude

Tables 2, 3 and 4. I suggest remove X2

Corrected in the text

Also in the OR you must include the p-value

Corrected in the text

Several paragraphs in this section are repetitive of the Results section.

Corrected in the text

I suggest to improve the discussion

Corrected in the text

Consider the title "Onychomycosis in two populations with different socioeconomic resources in an urban nucleus: A Cross-sectional study". 

Corrected in the text

Reviewer 2 Report (Previous Reviewer 1)

the article, with the limits also recognized by the authors, can be considered suitable for publication

Author Response

Dear Reviewer, thank you very much for considering our article for publication. 

With your considerations and improvements our article is more powerful.

Kind regards

Round 2

Reviewer 1 Report (Previous Reviewer 2)

No comments

This manuscript is a resubmission of an earlier submission. The following is a list of the peer review reports and author responses from that submission.

Round 1

Reviewer 1 Report

the authors present an interesting study on the prevalence of fungal nail infections in two groups of patients with different social conditions. the study is interesting. it certainly has two important problems: the first is the number of patients included in the study, for truly unique results there is certainly a need to expand the sample. another problem is related to the fact that no stratification of patients has been made according to the chronic pathologies that could indirectly modify the saprophytic flora

Reviewer 2 Report

In this study, they was isolate and identify onychomycosis-producing agents in homeless and compare them with agents isolated from populations attending podiatric clinics. This study is interesting but has several important limitations.

Major Comments

Methods

Design is a cross-sectional study (prevalence).

The main limitation was the small sample size (n=59) and convenience sample. This leads to a bias. How can you ensure that the sample is representative of the population?

Statistic analysis. Shapiro–Wilk test is used to test the normality of a data set, not to make comparisons.

Results

I don't think the two groups are comparable (40 group with home vs. 19 homeless group). In addition, the basic characteristics in both populations are not shown.

Tables 1, 2 and 3. They are only descriptive tables. Shapiro–Wilk test is used to test the normality of a data set, not to make comparisons.  Statistical analyzes are inadequate. For comparison, the authors should perform bivariate and multivariate logistic regression analyzes with Odds ratio (ORc) and adjusted (ORa).

It would have been interesting to know the prevalence rates according to sociodemographic characteristics in the two groups.

Discussion

In the “Discussion” section I would have wished to see more information.

Authors should describe and discuss any limitations of their study.

Conclusion. I don’t think that this article contains enough robust data to evidence the statement made in "Conclusion" section.

Reviewer 3 Report

In the homeless population, there 235 is a predominance of non-dermatophyte fungi, which may be due to predisposing factors 236 such as unhealthy habits, from nutrition to hygiene, and immunosuppression. 

Reviewer 4 Report

The manuscript entitled “Impact of onychomycosis in two populations with different socioeconomic resources in an urban nucleus” is interesting but suffers from significant shortcomings.

Major remarks the authors need to address:

The fungal identification procedure is not detailed in the methods section.

The culture of a fungus from a nail sample cannot be taken as a proof of onychomycosis as a contamination cannot be excluded. The authors should use a composite onychomycosis definition criteria including: direct microscopy, histology, and culture findings for each patient. They should consider using dermatophyte-PCR, which is particularly efficient for the diagnosis of onychomycosis, in further studies.

The participants have been sampled in distinct facilities, which probably display distinct airborne fungal contamination levels that might, at least partially, explain the differences in fungal culture results. This potential bias has not been controlled in the study.

The way the authors present their data in the tables is incompatibles with their objectives. The statistical tests do not compare the homeless or not populations; which is the study’s main objective.

The tables should be organized so that each population features are detailed in a distinct column.

Minor remarks:

The authors should follow the genus species nomenclature (Genus species) throughout the manuscript.

Table 1: Candida parapsilosis and Aspergillus have been misspelled as Candida Parapsoliosis; Aspegillus.

Round 2

Reviewer 1 Report

the revisions made by the authors do not fully satisfy the requests

Reviewer 2 Report

Include OR results in abstract.

Remove the Chi square value "X2" from the tables. Just include the p-value.

In tables 2 and 4, the statistical comparisons must be between Housed vs. homeless groups. 

Reviewer 3 Report

Very interesting work!

Reviewer 4 Report

My remarks were mainly directed against the study design. I can understand that they could not be adequately addressed by the authors